

# WAVINESS OF THE SOUTHERN HEMISPHERE WINTERTIME POLAR AND SUBTROPICAL JETS

*by*

*Jonathan E. Martin[1] and Taylor Norton[2]*

[1]Department of Atmospheric and Oceanic Sciences
[2]Antarctic Meteorological Research Center
University of Wisconsin-Madison
Madison, WI 53706
USA

Corr. Author:   Jonathan E. Martin,  jemarti1@wisc.edu

31                                        ABSTRACT

The recently developed average latitudinal displacement (ALD) methodology is applied

to assess the waviness of the austral winter subtropical and polar jets using three different
reanalysis data sets.  As in the wintertime Northern Hemisphere, both jets in the Southern
Hemisphere have become systematically wavier over the time series and the waviness of each jet
evolves quite independently of the other during most cold seasons.  Also, like its Northern
Hemisphere equivalent, the Southern Hemisphere polar jet exhibits no trend in speed (though it
is notably slower) while its poleward shift is statistically significant.  In contrast to its Northern
Hemisphere counterpart, the austral subtropical jet has undergone both a systematic increase in
speed as well as a statistically significant poleward migration.  Composite differences between
the waviest and least wavy seasons for each species suggest that the Southern Hemisphere's
lower stratospheric  polar vortex is negatively impacted by unusually wavy tropopause-level jets
of either species.  These results are considered in the context of trends in the Southern Annular
Mode as well as the findings of other related studies.

**KEYWORDS**: Southern Hemisphere, winter, polar jet, subtropical jet, waviness

## 1. Introduction

Consideration of changes in the behavior of the tropopause-level jet streams in a warming world has been catalyzed by the construction of long-period reanalysis data sets over the past three decades (Kalnay et al, 1996; Kistler et al., 2001; Kobayashi et al. 2015; Copernicus Climate Change Services [CS3], 2017). Recent analyses employing these data sets (e.g. Archer and Caldiera, 2008; Barnes and Screen, 2015; Gallego et al. 2005; Manney and Hegglin, 2018; Peña-Ortiz et al. 2013; Vavrus et al. 2017), in tandem with a number of studies based upon climate model output (e.g. Barnes and Polvani, 2013; Lorenz and DeWeaver, 2007; Miller et al. 2006; Yin, 2005), have produced a consensus view that poleward displacement of both jets accompanies warming. Along with an interest in latitudinal position, nearly all of the aforementioned studies have also addressed either observed and/or forecasted changes in the speed of the jet streams.

In a recent paper Martin (2021) offered a feature-based analysis of the *waviness* of the tropopause-level polar and subtropical jets during Northern Hemisphere winter (DJF). The analysis proceeded from the results of Christenson et al. (2017) that identified the isentropic layers that house the two species of jets during NH winter. He found that 1) the polar jet (POLJ) has undergone a statistically significant poleward migration over the time series, not matched by the subtropical jet (STJ), and 2) neither jet species exhibited a trend in its speed. Additionally, the analysis showed that both jets have become systematically wavier over the last 6 decades.

By virtue of its land/sea distribution, enhanced lower tropospheric warming at high latitudes of the NH, known as Arctic amplification, has recently emerged as a prominent signal of climate change (e.g., Serreze et al. 2009; Screen and Simmonds, 2013: and references therein). Francis and Vavrus (2012) were among the first to propose that changes in the undulatory nature

of the jet stream might be linked to Arctic amplification.  This suggestion initiated a decade-long
debate on this issue (e.g. Barnes, 2013; Blackport and Screen, 2020; DiCapua and Coumou,
2016; Francis, 2017; Francis and Vavrus, 2015; Francis et al. 2018; Martineau et al. 2017, Screen
and Simmonds, 2013; Vavrus, 2018).  As noted by Martin (2021), at least some of the
controversy and attendant lack of consensus surrounding this question (Barnes and Polvani,
2015) was nourished by the absence of a robust method of assessing the waviness of the
tropopause-level jets.  The average latitudinal displacement (ALD) methodology introduced in
Martin (2021) (briefly described later) offers one possible remedy to this deficiency.

The principle mode of variability in the SH extratropical circulation is the Southern

Annular Mode (SAM, Limpasuvan and Hartmann, 1999; Gong and Wang, 1999; Thompson and
Wallace, 2000), a nearly zonally symmetric structure with coincident geopotential height
anomalies of opposite signs in Antarctica and the middle latitudes.  In the decades prior to 2000,
the SH jets have shifted poleward and the SAM has tended toward positive polarity (e.g. Fogt
and Marshall, 2020).  These coincident trends have been presumed to be a result of ozone
depletion.  As the ozone recovers in the SH, simulations suggest a reversal of this trend may be
forthcoming (WMO, 2022).  Spenberger et al. (2020) have questioned whether the associated jet
displacement also explains shifts in the storm tracks across the hemisphere.  Instead they suggest
that SAM can be interpreted as a measure of the degree of coupling (or decoupling) between
Antarctica and the southern mid-latitudes.

Recently, considerable attention has been devoted to interrogating the zonally

asymmetric component of SAM (e.g. Fan 2007; Silvestri and Vera, 2009; Fogt et al. 2012; Rosso
et al. 2018; Campetelli et al. 2022).  This asymmetric component is characterized by a wave-3
pattern (Goyal et al., 2021; Goyal et al., 2022; Campetelli et al. 2022) with maximum amplitude
at 250 hPa in the Pacific and may be determinative of the overall positive trend in the SAM over
the reanalysis era. Such a wave-3, tropopause-level signal is immediately suggestive of the
influence of the jets. These observations motivate consideration of direct measurement of the
waviness of the SH wintertime jets.

Despite a number of recent studies that consider aspects of the interannual variability of

the austral winter subtropical jet (e.g. Gillett et al., 2021; Maher et al. 2019), to our knowledge, a
study by Gallego et al. (2005) is the only one to consider direct measurement of the waviness of
the austral winter jets. They employed an objective method focused on identifying the
geostrophic streamline of maximum average velocity at 200 hPa (i.e. the jet core at that level) to
separately consider the behaviors of the STJ and POLJ. This method allowed consideration of
the jets as continuous features around the hemisphere and thus enabled a number of novel
analyses of their behavior and trends. With particular relevance to the present study, they
considered a zonal index computed as the difference between the maximum and minimum
latitude of the jet core (i.e. the streamline at the core of the jet) on each day. A similar metric,
termed DayMaxMin, was employed by Barnes (2013) in her consideration of the behavior of the
NH 500 hPa flow. Though insightful, such a metric does not comprehensively account for the
waviness created by the full collection of troughs and ridges around the hemisphere that
routinely characterizes the jets.

In this paper we apply the methodology of Martin (2021) to assess recent trends in the

waviness of the SH wintertime polar and subtropical jets. The method of identifying the austral
winter polar and subtropical jet locations in isentropic space is described in Section 2 along with
a description of the data sets used. Also included there is a short description of the method of
assessing waviness introduced in Martin (2021). In Section 3, elements of the long-term trend
and interannual variability of the waviness of the austral winter polar and subtropical jets are
presented along with differences between composites of the waviest and least wavy seasons for
each species.  A summary and conclusions are offered in Section 4.

**2.  Data and Methodology**

In the foregoing analysis, the zonal ($u$) and meridional ($v$) winds as well as temperature ($T$),

at 6 h intervals from three different reanalysis data sets are employed.  72 austral winters (JJA)
(1948-2019) of the National Centers for Environmental Prediction/National Center for
Atmospheric Research (NCEP/NCAR) reanalysis, at 17 isobaric levels to 10 hPa on a 2.5°
latitude-longitude grid (Kalnay et al., 1996; Kistlet et al., 2001) are used.  We employ 62 winters
(1958-2019) of the Japanese 55-year (JRA-55) reanalysis with data on 60 vertical levels up to 0.1
hPa on a horizontal grid mesh of ~55 km (Kobayashi et al., 2015).  Finally, the ERA5 reanalysis
data set on 137 vertical levels from the surface to 80 km with a grid spacing of 31 km covering
the period from 1979 to 2019 (Copernicus Climate Change Service [CS3], 2017) are used as
well.  The waviness of the jets is assessed in the context of understanding their relationships to
the horizontal gradient of potential vorticity (PV) in prescribed isentropic layers.  A similar
approach was taken with respect to the STJ in recent work by Maher et al. (2019).  The first step
in the present analysis involves identification of the isentropic layers that house the austral winter
jets.  This was accomplished empirically by identifying the isentropic level at which the
maximum wind speed was observed in each grid column (between 10 and 80°S) at each analysis
time in JJA over the 62-year time series of the JRA-55 data set.  The use of isentropic space here
differs from the insightful approach taken by Manney et al. (2017) and Manney and Hegglin
(2018) which employed separate latitude and elevation criteria to differentiate between the STJ
and the POLJ.  Of the three data sets employed in the present work, the JRA-55 was chosen for
this preliminary analysis step because both its length of time series as well as its horizontal and
vertical resolutions are between those characterizing the other two data sets employed here.
Following Koch et al. (2006) we only considered columns in which the integral average wind
speed exceeded 30 ms$^{-1}$ in the 100-400 hPa layer.  The resulting distribution is clearly tri-modal
with frequency maxima, and therefore separate jet features, approximately located in the 305-
320, 340-355, and 395-410K isentropic layers (Fig. 1a).  The latter isentropic layer appears in the
lower stratosphere and is associated with the austral polar night jet (PNJ), which, being located
*above* the tropopause, is not a focus of the present analysis.  Further separation of the STJ and
POLJ is achieved through reference to Fig. 2 of Gallego et al. (2005) which strongly implies that
the STJ sharply peaks near 30°S while the POLJ more broadly peaks around 50°S.  Accordingly,
we further constrained the analysis to latitude bins 0-40°S for the STJ and 40 to 65°S for the
POLJ.  With this additional refinement, the analysis identifies the STJ in the 340-355K isentropic
layer and the POLJ in the 310-325K isentropic layer (Fig. 1b).  Similar analyses of the other two
data sets (not shown) revealed the robustness of this result.  It is important to note that 53.8% of
all qualifying columns (to 380K) in the 0-40°S bin (STJ) were in the 340-355K layer while
46.8% of all qualifying columns in the 40-65°S bin (POLJ) were in the 310-325K layer
supporting the isentropic assignments for the two species mentioned previously.  It is
immediately apparent, consistent with prior analyses (e.g. Bals-Elsholz et al. 2001, Nakamura
and Shimpo 2004, Gallego et al. 2005), that the STJ is the dominant jet feature in the southern
winter.
The analysis method to be used here involves assessment of the circulation which
requires calculation of contour length.  As a result, fair comparison among the different data sets
requires adoption of a uniform grid spacing.  Consequently, all three data sets were bilinearly
interpolated onto isentropic surfaces at 5K intervals (from 280 to 380K) and 2.5° latitude-
longitude grid spacing using programs within the General Meteorological Analysis Package
(GEMPAK) (desJardins et al., 1991).  The average PV and average zonal and meridional wind
speeds in both the polar jet (310:325K) and subtropical jet (340:355K) layers were then
calculated from the four times daily data for each day in each of the three time series.
As reviewed in Martin (2021), consideration of the quasi-geostrophic potential vorticity
(QGPV), following Cunningham and Keyser (2004), demonstrates that local maxima in the
cross-flow gradient of QGPV are collocated with maxima in the geostrophic wind speed.  In the
Southern Hemisphere, the jets lie on the high PV edge of this PV gradient.  By searching through
daily average isertels from -0.5 to -5.0 at 0.1 PVU intervals (1 PVU = $10^{-6}$ $m^2$ K $kg^{-1}$ $s^{-1}$), the
analysis identifies a "core isertel" along which the circulation per unit length (i.e. average speed)
is maximized in the separate POLJ (310:325K) and STJ (340:355K) isentropic layers for every
day in each of the time series.  This core isertel is, by design, an analytical proxy for the jet core.
A glimpse into the fidelity of this method in identifying the meandering cores of the POLJ and
STJ jets is illustrated in Fig. 2.  In each case the objectively identified core isertel, in black, lies
very near, or at, the center of the analyzed isotach maxima around the hemisphere with
physically defensible exceptions.  For instance, the red dashed lines in Fig. 2b indicate portions
of the bold black line in Fig. 2d (i.e. the overlying STJ core) suggesting that those portions of the
isotach maxima in Fig. 2b that are somewhat removed from the POLJ core isertel are the lower
portions of the overlying STJ core. Similarly, an extensive isotach maxima region in Fig. 2d has
a blue dashed line, a portion of the bold black line in Fig. 2b, slicing through it. This region,
well poleward of the STJ core isertel, is clearly the upper portion of the underlying POLJ core.
Figure 3a shows the average latitude for the core isertels of each jet species from each of the
three reanalyses data sets used in the study. The analyses return essentially identical results for
the core isertel of the STJ and very nearly identical results for the POLJ. Superimposing the
NCEP-NCAR reanalysis' JJA average 200 hPa isotachs on top of the STJ core isertels (Fig. 3b)
illustrates the fact that the average core isertel accurately represents the axis of the average STJ.
The relationship is also strong between the POLJ core isertels and the 700 hPa average isotachs
(Fig. 3c).
The waviness of each jet is assessed by calculating a hemispheric average of the meridional
displacements of the core isertel from its equivalent latitude – the northern extent of a polar cap
whose area is equal to the area enclosed by the core isertel. This metric is referred to as the
average latitudinal displacement (ALD). The method does not require that the core isertel be the
same in both jet layers on a given day, nor that it be the same from day-to-day in a given jet
layer. Consequently, it is important to examine its distribution in each jet layer over the entire
time series. Figure 4 portrays the frequency of occurrence of the core isertels in both the STJ
and POLJ layers for each of the three time series. The STJ core isertels peak between -1.95 and -
2.1 PVU across the three data sets. Considering all three data sets, 81.5% of all JJA days exhibit
a core isertel between -1 and -3 PVU in the STJ layer. The POLJ distribution is shifted toward
higher PV values. Overall, 74.8% of JJA days had a core isertel between -1 and -3 PVU in the
POLJ layer. The frequency of occurrence in the several isertelic bins for each species of SH jet
match quite well with what Martin (2021) found for the NH wintertime jets, even when
accommodating for the different isentropic layer for the austral POLJ.

## 3. Analysis


      The JJA seasonal average latitudinal displacement (ALD) of each jet is calculated as a

92-day average of the daily ALD in each cold season.  The results are shown in Fig. 5.  It is

instantly clear that, as in the NH, the POLJ is wavier than the STJ and that both jets have become

systematically wavier over the 62-year JRA-55 time series with $p < 0.004$ for both time series (a

one-sided Student's $t$-test was employed).  Interestingly, the austral winter STJ is less wavy than

its NH counterpart but the waviness of both has increased identically at 0.005 deg/yr (0.0125

deg/yr for NCEP since 1958 and -0.001 deg/yr for ERA-5).  The winter POLJ in the SH is, on

the other hand, wavier than in the NH and is trending faster (0.017 versus 0.009 deg/yr; 0.023

deg/yr for NCEP since 1958 and 0.009 deg/yr for ERA-5) than its NH complement.  Daily time

series of the ALD of each jet can also be examined to determine the extent to which the waviness

of the two jets covaries.  Figure 6 illustrates the POLJ and STJ daily ALDs for 1999 from each of

the three data sets.  The low correlation between the waviness of the two species in this example

year represents the rule rather than the exception.  All told, more than 93% of the STJ and POLJ

ALD seasonal time series constructed for this study are correlated with magnitudes less than 0.3.

This result strongly suggests that the waviness of the two species evolves independently.

      By definition, the average wind speed along the chosen core isertel on any given day

represents the average jet speed for that species on that day.  Time series of seasonal average jet

core wind speeds for the wintertime STJ and POLJ in both hemispheres are shown in Fig. 7.  As

in the NH winter (Martin, 2021), the austral POLJ shows almost no trend in jet core speed and

the slight change is not statistically significant.  Notably, however, the SH POLJ is ~6 m s$^{-1}$

slower on average than its NH equivalent.  Aside from the fact that the NCEP reanalysis is quite
different from the JRA-55 until about 1970, the austral winter STJ exhibits a robust, and
statistically significant (p-value < 0.001), increase in speed over the JRA-55 time series – in clear
contrast to its NH counterpart.  It is also apparent that the SH STJ is slightly weaker but less
interannually variable than the NH STJ.

Another characteristic of interest that emerges directly from the ALD analysis method is

the daily value of the jet core's equivalent latitude which closely approximates its zonally
averaged position.  Consequently, it is straightforward to construct a time series of the seasonal
average equivalent latitudes of the two species of jets, shown in Fig. 8.  Again, as in the NH, the
poleward shift of the SH POLJ is occurring three times faster than that exhibited by the STJ.  In
contrast to the situation in the NH, however, the slight poleward displacement of the SH STJ *is*,
like that of the POLJ, statistically significant (*p*-values for the POLJ and STJ are <0.001 and
0.002, respectively).  It is interesting to note that while the SH STJ is located at a roughly similar
latitude as the NH STJ throughout the time series, the SH POLJ is ~4° further poleward during
winter than the NH POLJ.  Overall, a much more systematic and dramatic poleward migration of
the two jets has occurred over the last 6 decades in SH winter as compared to NH winter.

Next we consider aspects of the analysis in the context of the SAM.  Figure 9 shows a

histogram of the JJA average SAM index (calculated after Gong and Wang (1999))
superimposed upon the average JJA ALD from the JRA-55 reanalysis.  The tendency toward
positive SAM over the time series appears to be reflected in the increase in ALD.  However, the
correlation between the two time series is 0.053 suggesting almost no relationship exists between
the two.
In order to investigate the relationship of ALD to extremes in the polarity of the SAM
index, the three winter months with the most positive and most negative SAM extremes since
1979 were considered.  The core isertels of the POLJ (from the JRA-55 reanalysis) for each of
these three months is portrayed in Fig. 10.  Positive extremes of SAM (Figs. 10a, c, and e) show
a clear poleward encroachment of the SH polar jet while negative extremes (Figs. 10b, d, and f)
suggest the opposite.  There appears to be no systematic connection, however, between extremes
in SAM and the waviness of the POLJ as quantified by ALD.
Thus far the analysis has presented elements of the seasonal average behavior of the
austral winter jet species.  The methodology, of course, allows for evaluation of daily time series
of ALD as well and, in fact, such an analysis underlies the presentation in Fig. 6.  Using such
daily time series, identification of the waviest and least wavy seasons for each jet species since
1979 is accomplished by summing the daily departures from calendar-day average ALD over the
92 days of each cold season.  The list of such seasonally integrated departures from average
waviness for each species of jet for each reanalysis data set is shown in Table 1.  From this list,
the 5 waviest and 5 least wavy seasons for each jet species were selected to construct composites
of geopotential height at several isobaric levels employing the JRA-55 data.  In the foregoing
analysis, height differences are obtained by subtracting values associated with the composite
least wavy seasons from those associated with the composite waviest seasons.
Figure 11a shows the 500 hPa geopotential height differences between the waviest and
least wavy POLJ seasons.  Wavy POLJ years are characterized by positive height anomalies over
the continent and adjacent to its east and west coasts with belts of negative anomalies in a
crescent stretching from southwest of Chile and then extending from the east coast of South
America to southern Africa toward Australia, suggestive of a negative SAM.  The strongest
negative height anomalies in such seasons occur west of South Africa implying a slight
weakening of the zonal winds just south of the Cape of Good Hope.  Meanwhile wavy STJ years
exhibit negative composite height differences in roughly the same locations as the positive
composite differences just described for wavy POLJ years (Fig. 11b), suggestive of a positive
SAM.  These composite difference patterns strengthen slightly at 250 hPa (Fig. 12) suggesting an
equivalent barotropic structure to the tropospheric portion of the difference fields.

The difference fields at 50 hPa imply that the waviness of both jets exerts an

influence on the strength of the austral polar vortex in the lower stratosphere.  The anomalous
height field associated with wavy POLJ years (Fig. 13a) suggests a broad, though modest,
anticyclonic circulation anomaly just off the pole in the Western Hemisphere.  Such a
perturbation flow would appear to interfere with the establishment and/or persistence of strong
vortex flow in the same location.  Wavy STJ seasons also impose a dipole of positive heights the
axis of which stretches from Cape Horn to East Antarctica (Fig 13b).  Such a configuration
implies that the polar vortex is both weaker and displaced off the pole in winters with wavy
STJs.  Thus, the analysis suggests that in winters characterized by unusually wavy jets of either
species, the SH polar vortex is likely weaker than normal.  Further investigation of this intriguing
implication is the subject of ongoing work.

**4. Summary**

The analysis presented here extends the application of a method developed by Martin

(2021) to assess the waviness of the tropopause-level jets to analysis of the austral winter polar
and subtropical jets.  The analysis demonstrates that both jets have become systematically wavier
over the past 60+ years.  In addition, as in the NH, the waviness of the two species of austral
winter jets is largely uncorrelated suggesting little systematic influence of one on the other
throughout the season.  Along with these similarities, there appear to be some fundamental
asymmetries in the behavior of the wintertime tropopause-level jets between the hemispheres.
The austral POLJ, like its NH counterpart, has exhibited no trend in its average speed over the
time series, though it is notably slower than its NH wintertime equivalent.  The STJ, on the other
hand, has roughly the same speed as that in the NH winter but, unlike its NH counterpart, has
undergone a systematic, statistically significant increase in its core speed since ~1960.
Additionally, as opposed to the situation in the NH where only the POLJ migration toward to
pole is statistically significant, **both** SH jets exhibit a significant poleward creep with the POLJ
encroachment occurring at ~3x the rate of that characterizing the STJ.

The observed poleward migration of the STJ reported here is consistent with the analysis

of CMIP5 simulations of historical and projected changes to the SH wintertime STJ by Chenoli
et al. (2017).  Though the present work employs a similarly dynamical definition of the STJ as
that used in the study by Maher et al. (2019), they found no evidence of a poleward shift of the
SH wintertime STJ.  We suggest that the emphasis on empirically identifying a core isertel,
rather than the maximum gradient of θ on a predetermined isertelic surface (i.e. 2 PVU as the
dynamic tropopause) may account for this difference.

Finally, circulation differences between the waviest and least wavy POLJ and STJ

seasons are manifest in both the troposphere and lower stratosphere.  In the troposphere the
signals are not as coherent in the SH as they were revealed to be in the NH (Martin 2021).
Interestingly, the analysis implies that when either the POLJ or STJ is wavier than normal in a
given winter, the lower stratospheric polar vortex is negatively impacted. Again, this is different
from the behavior of the NH polar vortex in the face of extremes in waviness.
The results presented here, combined with those in Martin (2021), demonstrate that in
both hemispheres a wavier than normal STJ during winter serves to weaken the lower
stratospheric polar vortex. Though, as suggested by the analysis supporting Fig. 6, the STJ and
POLJ do not appear to influence one another systematically, there are still instances in which the
waviness of the two jets can be phased so as to promote intense interactions. Daily perusal of
hemispheric synoptic maps suggests that such instances of jet interaction often lead to intense
lower tropospheric cyclogenesis events. Current research is examining whether such jet
interaction-induced cyclogenesis events from specific seasons systematically correspond to
episodes of polar vortex weakening.

COMPETING INTERESTS: The contact author has declared that none of the authors has any
competing interests.

AUTHOR CONTRIBUTIONS: J. Martin completed the ALD analysis and did all the writing,
figure drafting and preparation of the manuscript for submission. T. Norton performed the
analysis that determined the POLJ and STJ isentropic housings during SH winter.

ACKNOWLEDGEMENTS: This work was supported by the National Science Foundation under
grants ATM-1640055 and NSF-2055667. JRA-55 data available from the Research Data
Archive at the National Center for Atmospheric Research. The authors would like to thank Prof.
Andrea A. Lopez-Lang for helpful comments and suggestions.

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

|  | **_POLJ_** | | | **_STJ_** | | |
|---|---|---|---|---|---|---|
|  | _NCEP_ | _JRA-55_ | _ERA5_ | _NCEP_ | _JRA-55_ | _ERA5_ |
| 1979 | _-45.416403_ | _-19.684881_ | _-64.232707_ | -3.7754167 | 11.1345645 | 0.70639878 |
| 1980 | _-59.380403_ | _-58.393881_ | _-63.657707_ | -3.2564167 | 4.47656452 | -0.8576012 |
| 1981 | 18.8845972 | 36.4021194 | 21.8872927 | -4.4154167 | 5.17956452 | -2.2326012 |
| 1982 | -24.813403 | 3.63785707 | -15.198707 | 41.2785833 | 16.1355645 | 10.5773988 |
| 1983 | -16.281403 | 35.8650731 | -15.658707 | _-18.131417_ | _-10.037435_ | _-21.992601_ |
| 1984 | -14.954403 | 4.06711936 | -6.9887073 | 15.8335833 | 19.8715645 | 11.5133988 |
| 1985 | 4.02659722 | 10.3371194 | -20.535707 | **54.3615833** | **38.7185645** | **21.4393988** |
| 1986 | 24.2525972 | 43.6271194 | 20.5902927 | -6.9904167 | 0.00256452 | -10.285601 |
| 1987 | 62.9565972 | 77.0631194 | 16.8692927 | -8.9194167 | 0.57256452 | -9.2566012 |
| 1988 | -2.5614028 | -7.4278806 | -35.518707 | -1.9554167 | 2.34556452 | -5.2936012 |
| 1989 | -33.646403 | 9.93808658 | -16.752707 | 35.2235833 | 29.6575645 | 19.4843988 |
| 1990 | 21.8045972 | 40.1761194 | 5.93129268 | 28.6225833 | 8.32356452 | -0.0366012 |
| 1991 | **98.1615972** | **104.846187** | **79.9922927** | 15.9005833 | 13.6185645 | 10.0163988 |
| 1992 | _-31.301403_ | _-26.480881_ | _-43.682707_ | 23.9145833 | 30.1255645 | 22.5733988 |
| 1993 | 45.9685972 | 64.4221194 | 23.6692927 | -5.4784167 | 7.54756452 | -0.9296012 |
| 1994 | _-29.454403_ | _-32.656881_ | _-69.886707_ | **51.5895833** | **30.9815645** | **21.4813988** |
| 1995 | -22.226403 | -20.908881 | -47.179707 | _-5.1054167_ | _-12.989435_ | _-16.721601_ |
| 1996 | **80.0555972** | **96.1361194** | **86.8222927** | -2.3444167 | -10.092435 | -11.395601 |
| 1997 | **68.8895972** | 57.6655297 | **78.5282927** | 2.23058333 | -8.3644355 | -11.693601 |
| 1998 | _-27.166403_ | _-32.68934_ | _-70.988707_ | 18.5915833 | -2.3754355 | -7.9706012 |
| 1999 | 36.1115972 | -22.593881 | -44.562707 | _3.60158333_ | _-23.970435_ | _-32.015601_ |
| 2000 | 57.1715972 | 17.3883325 | 16.0832927 | 49.9395833 | 18.8905645 | 12.2183988 |
| 2001 | 51.6315972 | 26.2991194 | 8.28429268 | 46.9905833 | 7.20656452 | 1.48939878 |
| 2002 | 30.0675972 | 35.9181194 | 21.4212927 | **65.2545833** | **47.0115645** | **40.0813988** |
| 2003 | 70.6935972 | 52.1291194 | 24.5692927 | 12.5915833 | -3.7804355 | -11.507601 |
| 2004 | 27.8395972 | -18.835881 | -31.660707 | 39.5535833 | 19.0855645 | 13.4163988 |
| 2005 | 48.0095972 | 26.0351194 | -2.9987073 | _-10.510417_ | _-21.297435_ | _-26.212601_ |
| 2006 | 76.9665972 | 27.7838267 | 24.9342927 | 29.3135833 | -2.1904355 | -10.139601 |
| 2007 | 60.9595972 | 55.4256292 | 46.9952927 | 38.6865833 | 17.2975645 | 14.1103988 |
| 2008 | **67.6425972** | **67.2851194** | **66.7882927** | _-4.0874167_ | _-21.790435_ | _-25.102601_ |
| 2009 | 69.9215972 | 17.7955696 | 23.8622927 | 22.6285833 | -4.6854355 | -8.0676012 |
| 2010 | 41.5965972 | 13.4191194 | 3.93329268 | 31.9945833 | 16.0065645 | 11.1233988 |
| 2011 | **118.932597** | **111.764119** | **79.1722927** | 11.7745833 | -5.6934355 | -8.7496012 |
| 2012 | 38.3955972 | 9.84011936 | -2.5287073 | 54.8005833 | 14.8235645 | -1.2216012 |
| 2013 | 32.3355972 | -0.7048806 | -14.266707 | **67.4165833** | **25.3645645** | **13.6133988** |
| 2014 | 52.2325972 | 45.4011194 | -60.736707 | 40.1415833 | 20.9895645 | 6.32532378 |
| 2015 | 65.0135972 | 38.0481194 | 18.8882927 | 14.6575833 | 1.69656452 | -1.7356012 |
| 2016 | 51.9375972 | 19.3210046 | 15.3602927 | 22.3815833 | 2.71556452 | -0.3676012 |
| 2017 | 15.4975972 | -14.224881 | -38.558707 | 30.2145833 | 2.97356452 | -2.2008762 |
| 2018 | 70.8755972 | 21.0891194 | 3.86429268 | 3.15258333 | -7.7994355 | -11.277601 |
| 2019 | 68.5365972 | 5.97811936 | -22.852707 | **58.1465833** | **21.7315645** | **7.09439878** |

TABLE 1 Integrated seasonal departures from average ALD (degrees) for polar and subtropical jets from the three reanalysis data sets employed in this work. Bold (underlined italics) represents one of the top 5 waviest (least wavy) seasons.



                                                   FIGURE CAPTIONS


Fig. 1 (a) Distribution of grid-column maximum wind speeds found in 5K isentropic layers from
10 - 80°S for every 6h analysis time in JJA from 1958-2019 from the JRA-55 reanalysis. (b)  As
for Fig. 1a except limited to (i) grid-columns in which the integral average wind speed from 400
to 100 hPa exceeded 30 m s$^{-1}$ and (ii) to latitudes 0 - 40°S for the STJ and (iii) latitudes 40 to
65°S for the POLJ.

Fig. 2 (a) Isotachs of the daily averaged wind speed (contoured every 10 m s$^{-1}$ and shaded above
30 m s$^{-1}$) and the core isertel (bold black line) in the 310:325K isentropic layer on 13 July 1995
from the JRA-55 reanalysis data.  The core isertel value is -1.3 PVU.  (b) As in (a) but for 24
August 2001.  Core isertel value is -2.0 PVU.  Dashed red line indicates portion of the core
isertel from the overlying STJ layer (depicted in Fig. 2d).  (c) As in (a) but for wind speeds and
core isertel in the 340:355K isentropic layer on 13 July 1995.  Core isertel value is -3.6 PVU.  (d)
As in (c) but for 24 August 2001.  Core isertel value is -1.4 PVU.  Dashed blue line indicates a
portion of the core isertel from the underlying POLJ layer (depicted in Fig. 2b).  See text for
further explanation.

Fig. 3 (a) Solid (dashed) lines are the positions of the average core isertels of the STJ (POLJ)
from each of the three reanalyses employed in this study.  The different reanalyses are color
coded.  (b) Thick solid lines are the positions of the average core isertels for the STJ from each
of the reanalyses superimposed with JJA average 200 hPa isotachs from the NCEP-NCAR
reanalysis.  (c) Thick dashed lines are the positions of the average core isertels for the POLJ
superimposed with JJA average 700 hPa isotachs from the NCEP-NCAR reanalysis.

Fig. 4  Frequency of occurrence of the core isertel value for each reanalysis time series in (a) the
STJ layer and (b) the POLJ layer.  Solid blue, red and green lines in (a) and (b) are the SH
distributions from the NCEP, JRA55 and ERA5, respectively.  The dashed blue, red and green
lines are the NH distributions from the NCEP, JRA55 and ERA5 reanalyses, respectively.  In (b),
the NH distributions are from the 315:330K layer which houses the POLJ in the boreal winter.
Thin blue, red and green lines in (a) and (b) indicate the peak values of the core isertel in each
layer from each data set.  Isertel values are given in potential vorticity units (PVU,
$1\ PVU = 10^6\ K\ m^2\ kg^{-1}\ s^{-1}$), and are multiplied by -1 for the NH values.

Fig. 5 Seasonal average ALD (in degrees) of the SH wintertime subtropical and polar jets for
each cold season in the three reanalysis time series.  The polar jet values are in the three shades
of blue while the subtropical jet values are in the three three shades of red.  The dashed black line
through each time series represents the trend line for each (derived from the JRA-55 time series)
and is significant at the 96% level.  Gray lines are the boreal winter ALD analysis from Fig. 6 of
Martin (2021).The "YEAR" on the abscissa indicates the year in which December of that cold
season occurred.

Fig. 6  Time series of the daily ALD of the polar (blue lines) and subtropical (red lines) jets from
the (a) NCEP-Reanalysis, (b) JRA-55, and (c) ERA5 data sets for austral winter 1999.  The
correlation between the two times series from each data set is indicated.

Fig. 7  Seasonal average **U** along the core isertel for the subtropical (red lines) and polar (blue
lines) jets from each of the three SH reanalysis data sets.  The thin black lines are trend lines for
each time series from the JRA-55 data.  Gray lines are the equivalent boreal winter **U** analysis
from Fig. 9 of Martin (2021).

Fig. 8  Time series of the seasonal average equivalent latitude of the polar (blue lines) and
subtropical (red lines) jets from the three different SH reanalysis data sets.  The thin black lines
are the trend lines (from the JRA-55 data) and are significant above the 99% leve for both jet
species.  Gray lines are the boreal winter equivalent latitude analysis from Fig. 10 of Martin

574    (2021).


Fig. 9 JJA average SAM index (histogram) from NCEP's Climate Prediction Center.  The index
is calculated by projecting the daily 700 hPa geopotential height anomalies poleward of 20S onto
the leading pattern of the Antarctic Oscillation (AAO) ofGong and Wang (1999).  Black solid
line is theJJA average ALD of the POLJ from the JRA-55 reanalysis.

Fig. 10 Spaghetti plots of core isertels from SH summer months with maximum positive (red)
and negative (blue) SAM indices since 1979. (a) Daily JRA-55 core isertels from June 2009, the
June with the most positive SAM in the record.  (b) As for Fig. 10a but for June 1992, the June
with the most negative SAM in the record.  (c) As for Fig. 10a but for July 1998.  (d) As for Fig.
10b but for July 1995.  (e) As for Fig. 10a but for August 1994.  (f) As for Fig. 10b but for
August 1981.  Average ALD for the given months are listed in the bottom left of each panel.

Fig. 11  500 hPa height differences between the composite waviest and least wavy (a) polar jet
and  (b) subtropical jet seasons constructed from the JRA-55 reanalysis.  See Table 1 for
identification of the specific years comprising each composite.  Positive (negative) height
differences are in solid red (blue) lines labeled in m and contoured every 10 m (-10 m) beginning
at 10 m (-10 m).

Fig. 12  250 hPa height differences between the composite waviest and least wavy (a) polar jet
and  (b) subtropical jet seasons constructed from the JRA-55 reanalysis.  See Table 1 for
identification of the specific years comprising each composite.  Positive (negative) height
differences are in solid red (blue) lines labeled in m and contoured every 10 m (-10 m) beginning
at 10 m (-10 m).

Fig. 13  50 hPa height differences between the composite waviest and least wavy (a) polar jet
and  (b) subtropical jet seasons constructed from the JRA-55 reanalysis.  See Table 1 for
identification of the specific years comprising each composite.  Positive (negative) height
differences are in solid red (blue) lines labeled in m and contoured every 10 m (-10 m) beginning
at 10 m (-10 m).


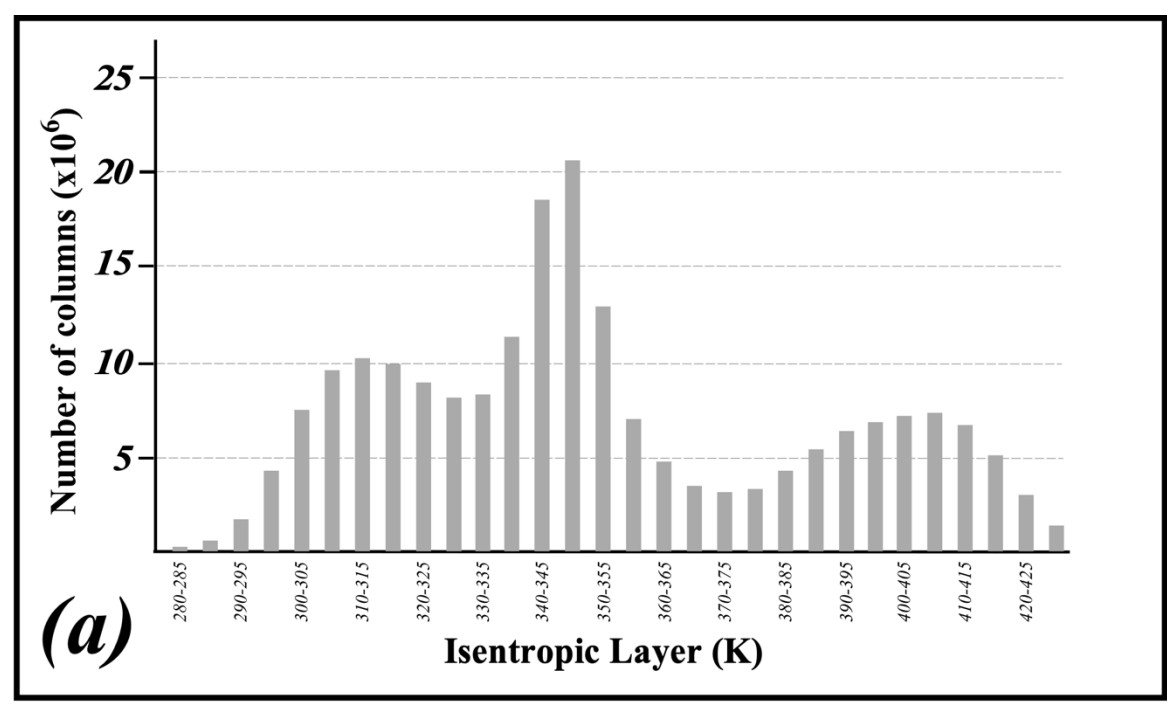

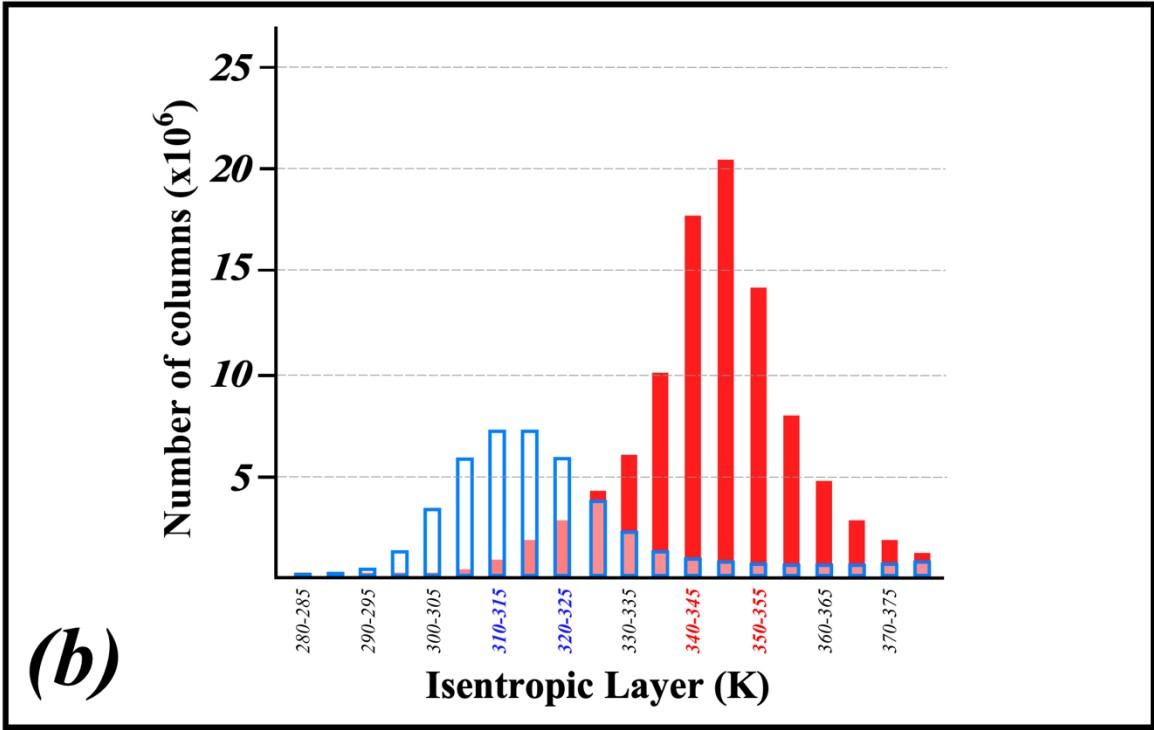

Fig. 1 (a) Distribution of grid-column maximum wind speeds found in 5K isentropic layers from 10 - 80°S for every 6h analysis time in JJA from 1958-2019 from the JRA-55 reanalysis. (b) As for Fig. 1a except limited to (i) grid-columns in which the integral average wind speed from 400 to 100 hPa exceeded 30 m s⁻¹ and (ii) to latitudes 0 - 40°S for the STJ and (iii) latitudes 40 to 65°S for the POLJ.


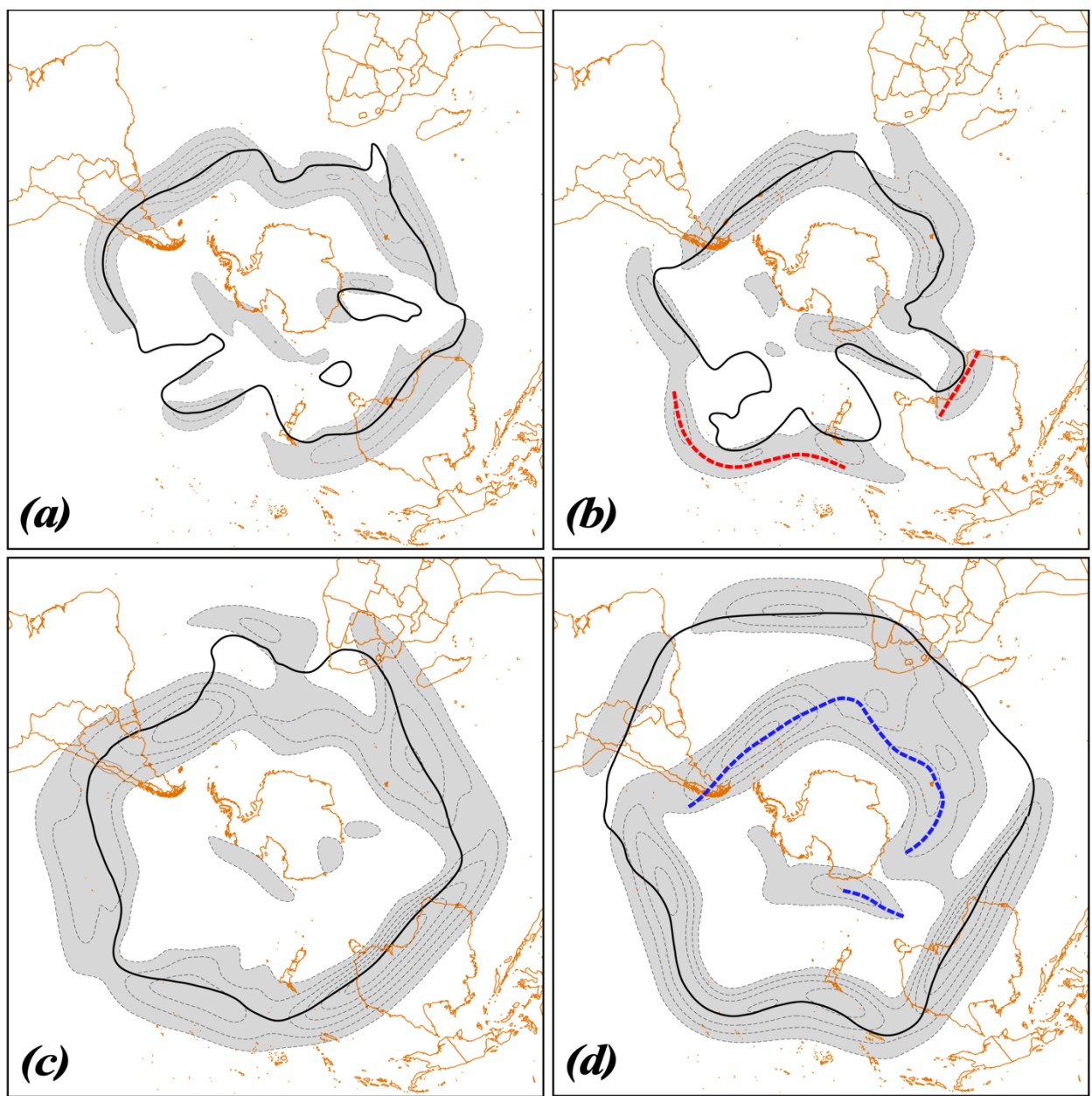

Fig. 2 (a) Isotachs of the daily averaged wind speed (contoured every 10 ma-1 and shaded above 30 m s-1) and the core isertel (bold black line) in the 310:325K isentropic layer on 13 July 1995 from the JRA-55 reanalysis data. The core isertel value is -1.3 PVU. (b) As in (a) but for 24 August 2001. Core isertel value is -2.0 PVU. Dashed red line indicates portion of the core isertel from the overlying STJ layer (depicted in Fig. 2d). (c) As in (a) but for wind speeds and core isertel in the 340:355K isentropic layer on 13 July 1995. Core isertel value is -3.6 PVU. (d) As in (c) but for 24 August 2001. Core isertel value is -1.4 PVU. Dashed blue line indicates a portion of the core isertel from the underlying POLJ layer (depicted in Fig. 2b). See text for further explanation.



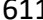

Fig. 3 (a) Solid (dashed) lines are the positions of the average core isertels of the STJ (POLJ) from each of the three reanalyses employed in this study. The different reanalyses are color coded. (b) Thick solid lines are the positions of the average core isertels for the STJ from each of the reanalyses superimposed with JJA average 200 hPa isotachs from the NCEP-NCAR reanalysis. (c) Thick dashed llines are the positions of the average core isertels for the POLJ superimposed with JJA average 700 hPa isotachs from the NCEP-NCAR reanalysis.

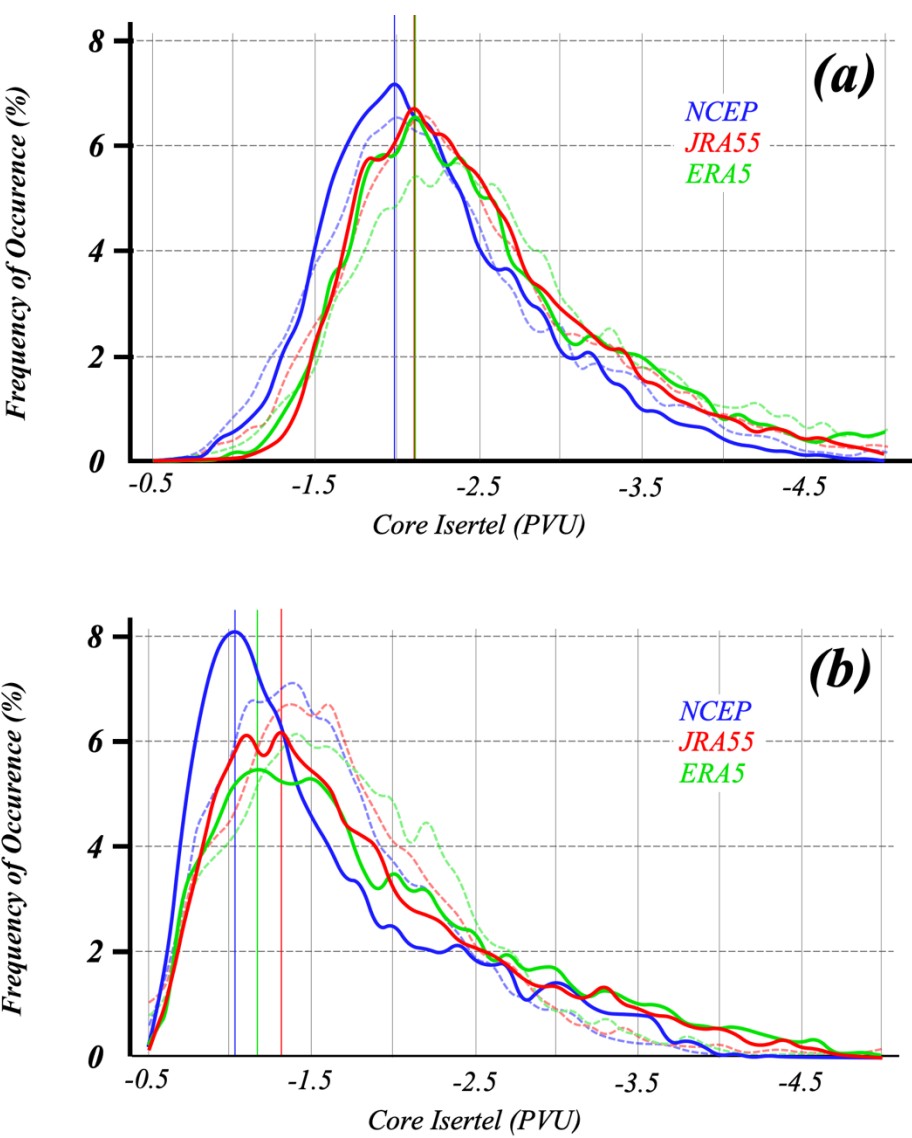

Fig. 4 Frequency of occurrence of the core isertel value for each reanalysis time series in (a) the STJ layer and (b) the POLJ layer.  Solid blue, red and green lines in (a) and (b) are the SH distributions from the NCEP, JRA55 and ERA5, respectively.  The dashed blue, red and green lines are the NH distributions from the NCEP, JRA55 and ERA5 reanalyses, respectively.  In (b), the NH distributions are from the 315:330K layer which houses the POLJ in the boreal winter.  Thin blue, red and green lines in (a) and (b) indicate the peak values of the core isertel in each layer from each data set.  Isertel values are given in potential vorticity units (PVU, 1 PVU = $10^6$ K m$^2$ kg$^{-1}$ s$^{-1}$) and are multplied by -1 for the NH values.



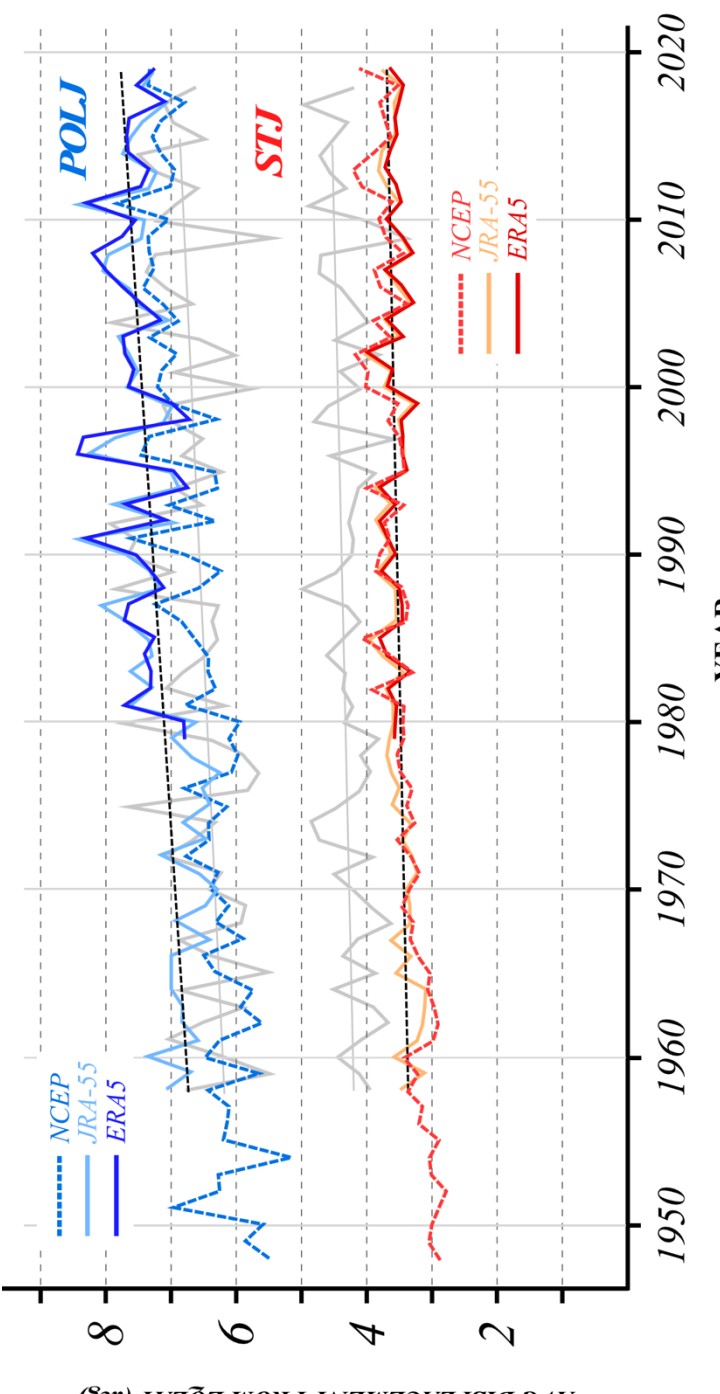

Fig. 5 Seasonal average ALD (in degrees) of the SH wintertime subtropical and polar jets for each cold season in the three reanalysis time series. The polar jet values are in the three shades of blue while the subtropical jet values are in the three three shades of red. The dashed black line through each time series represents the trend line for each (derived from the JRA-55 time series) and is significant at the 96% level. Gray line is the boreal winter average ALD from 1958 onward portrayed in Fig. 6 of Martin (2021).The "YEAR" on the abscissa indicates the year in which December of that cold season occurred.

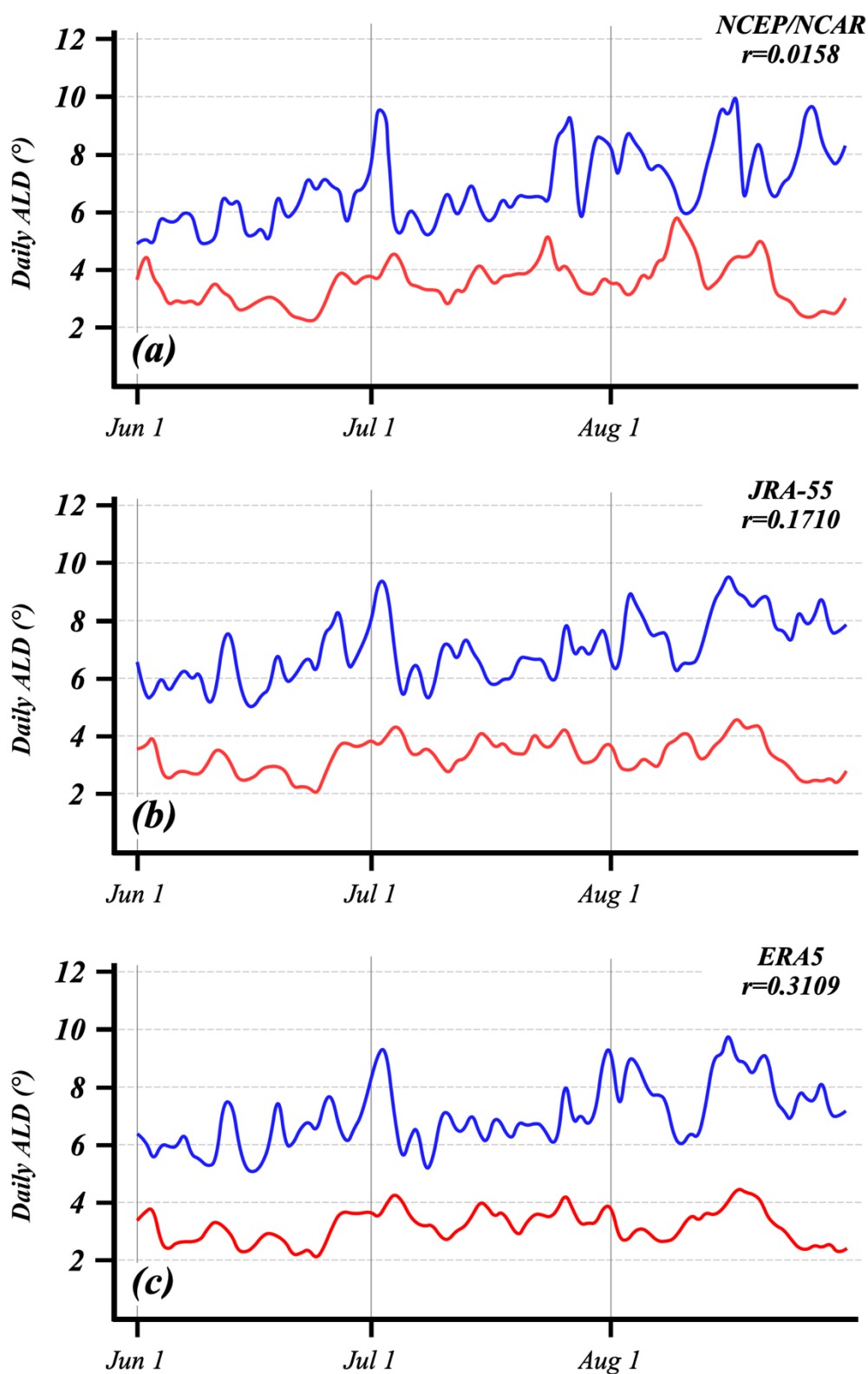

Fig. 6 Time series of the daily ALD of the polar (blue lines) and subtropical (red lines) jets from the (a) NCEP-Reanalysis, (b) JRA-55, and (c) ERA5 data sets for austral winter 1999. The correlation between the two times series from each data set is indicated.




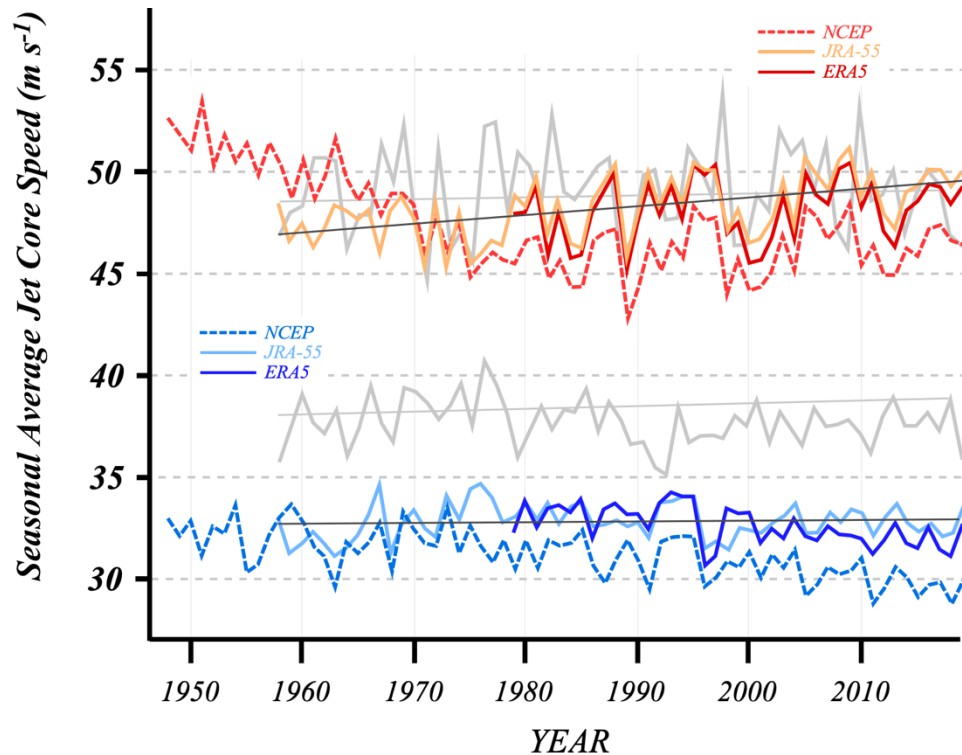

Fig. 7  Seasonal average **U** along the core isertel for the subtropical (red lines) and polar (blue lines) jets from each of the three SH reanalysis data sets.  The thin black lines are trend lines for each time series from the JRA-55 data.  Gray line is the average (1958-2018) boreal winter **U** analysis for each jet from the three data sets in Fig. 9 of Martin (2021).



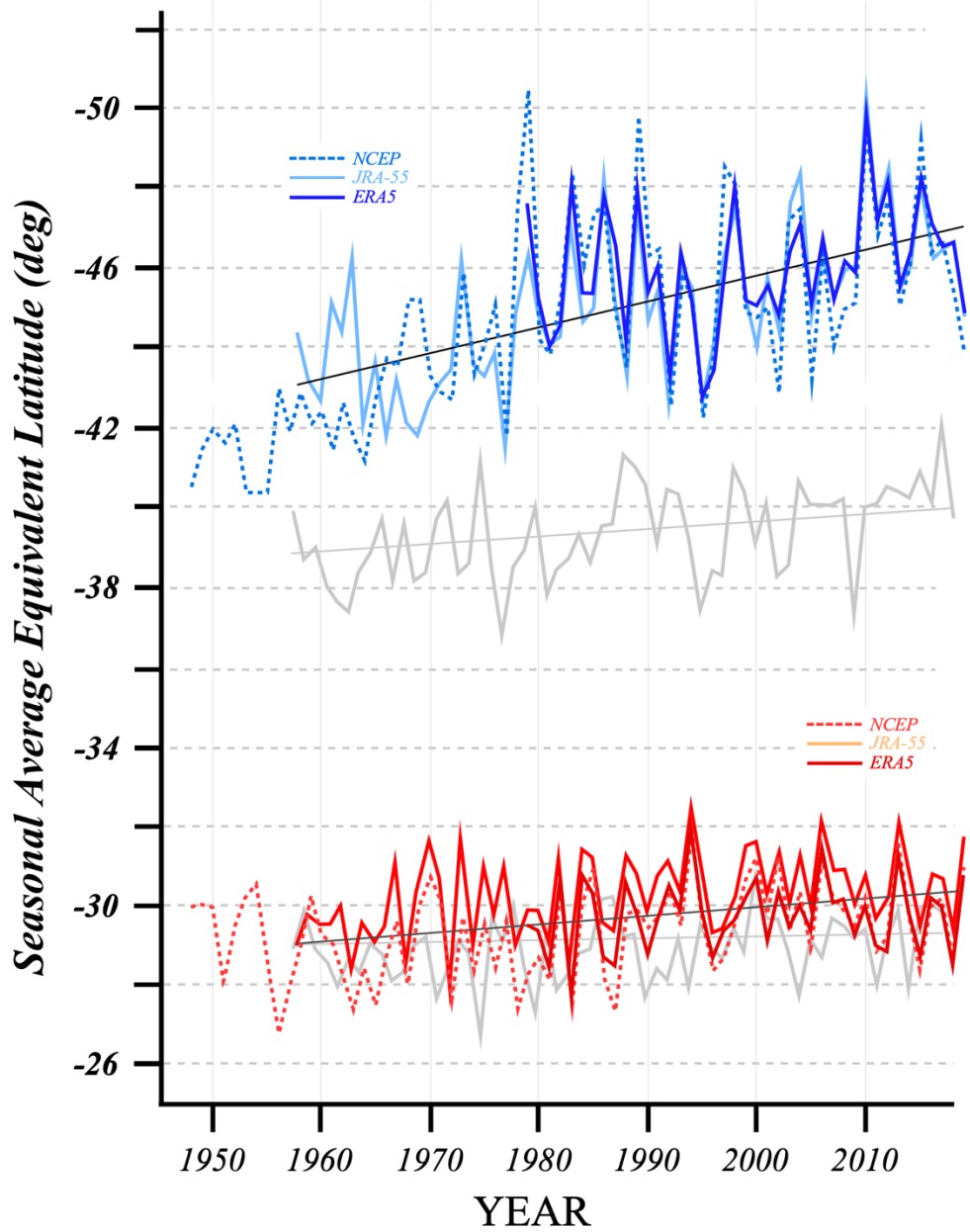

Fig. 8  Time series of the seasonal average equivalent latitude of the polar (blue lines) and subtropical (red lines) jets from the three different SH reanalysis data sets.  The thin black lines are the trend lines (from the JRA-55 data) and are significant abov e the 99% level for both jet species.  Gray line is the boreal winter average (1958-2017) equivalent latitude for each jet from the three reanalysis data sets portrayed in Fig. 10 of Martin (2021).


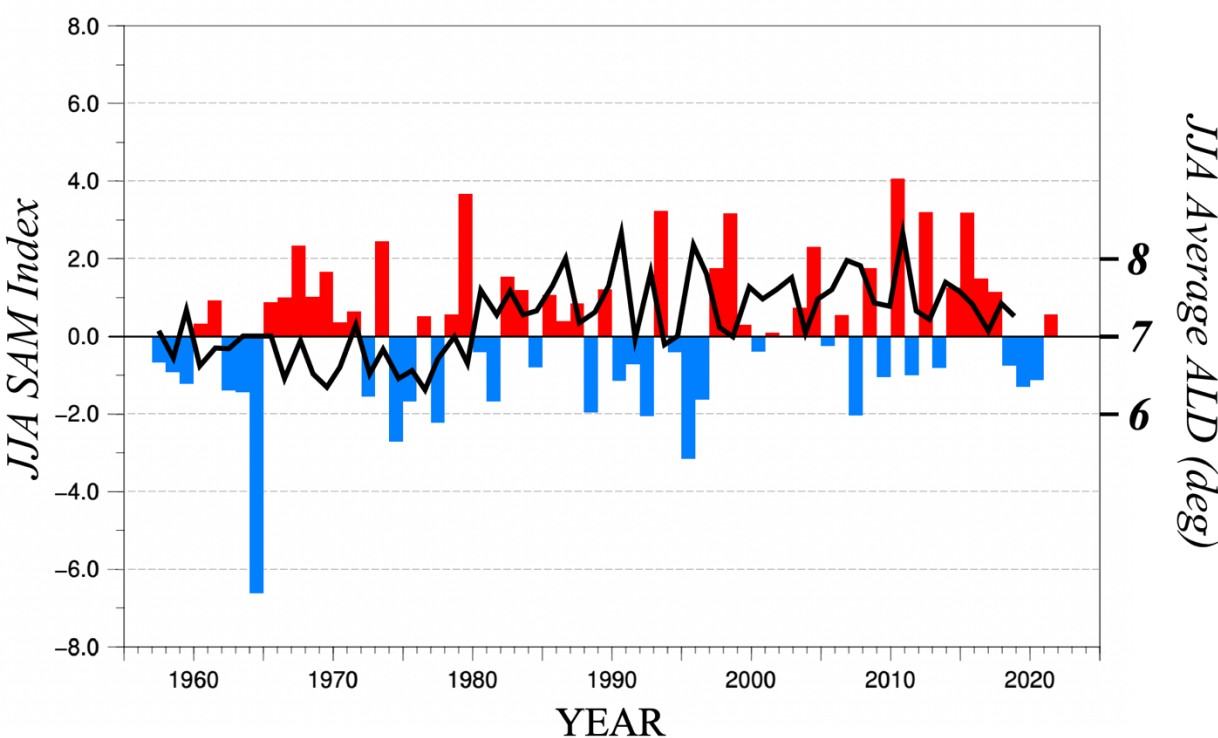

Fig. 9 JJA average SAM index (histogram) from NCEP's Climate Prediction Center. The index is calculated by projecting the daily 700 hPa geopotential height anomalies poleward of 20S onto the leading pattern of the Antarctic Oscillation (AAO) ofGong and Wang (1999). Black solid line is the JJA average ALD of the POLJ from the JRA-55 reanalysis.


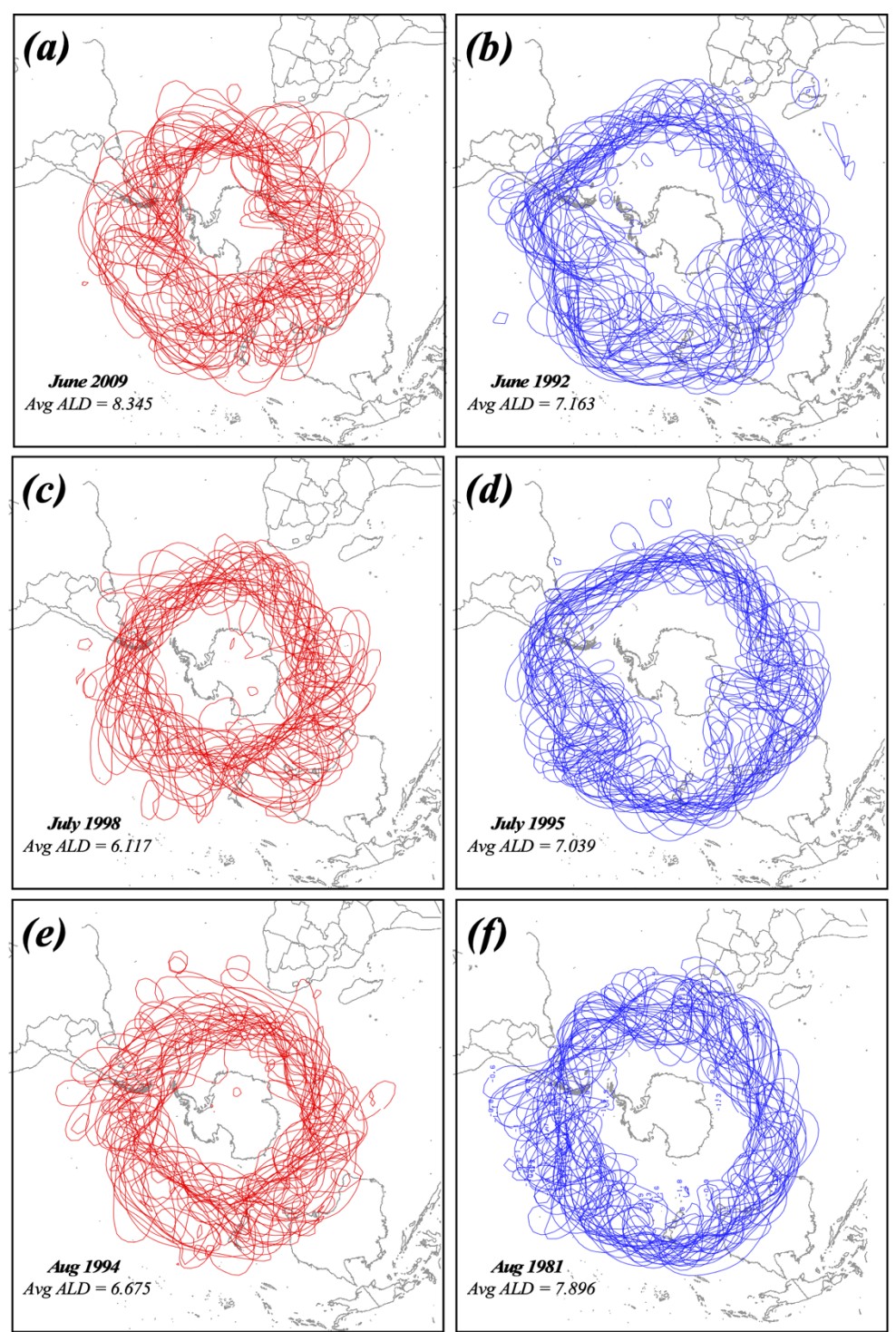

Fig. 10 Spaghetti plots of core isertels from SH summer months with maximum positive (red) and negative (blue) SAM indices since 1979. (a) Daily JRA-55 core isertels from June 2009, the June with the most postive SAM in the record. (b) As for Fig. 10a but for June 1992, the June with themost negative SAM in the record. (c) As for Fig. 10a but for July 1998. (d) As for Fig. 10b but for July 1995. (e) As for Fig. 10a but for August 1994.(f) As for Fig. 10b but for August 1981. Average ALD for the given months are listed in the bottom left of each panel.


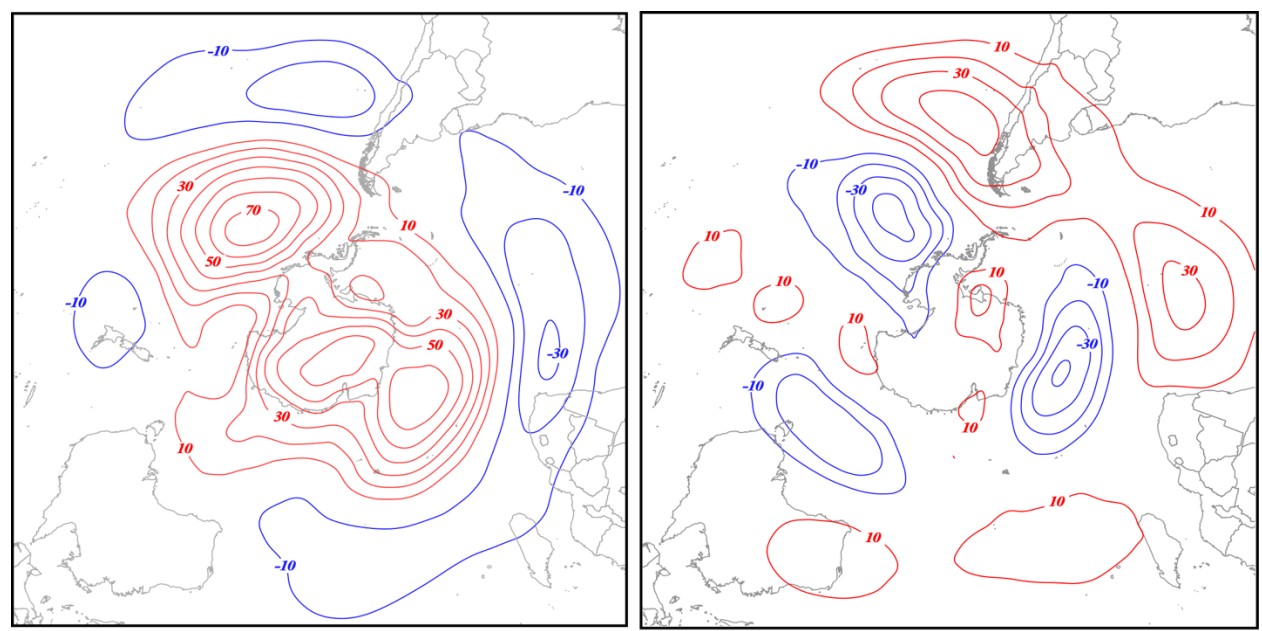

Fig. 11 500 hPa height differences between the composite waviest and least wavy (a) polar jet and (b) subtropical jet seasons constructed from the JRA-55 reanalysis. See Table 1 for identification of the specific years comprising each composite. Positive (negative) height differences are in solid red (blue) lines labeled in m and contoured every 10 m (-10 m) beginning at 10 m (-10 m).





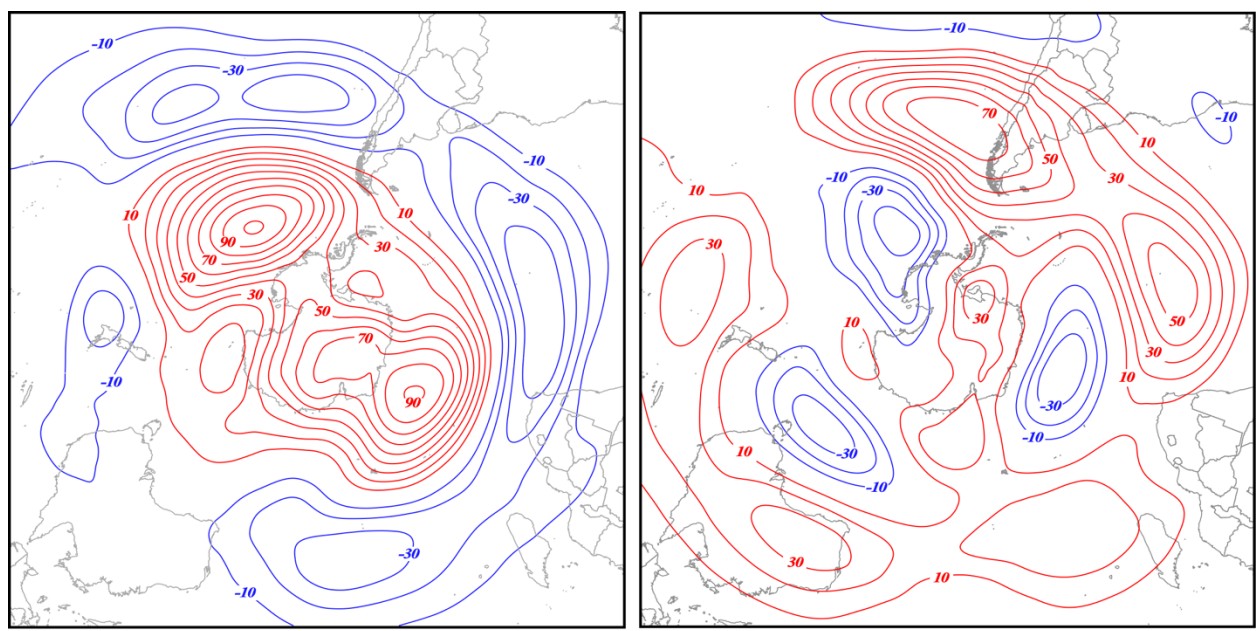

Fig. 12  250 hPa height differences between the composite waviest and least wavy (a) polar jet and (b) subtropical jet seasons constructed from the JRA-55 reanalysis.  See Table 1 for identification of the specific years comprising each composite.  Positive (negative) height differences are in solid red (blue) lines labeled in m and contoured every 10 m (-10 m) beginning at 10 m (-10 m).





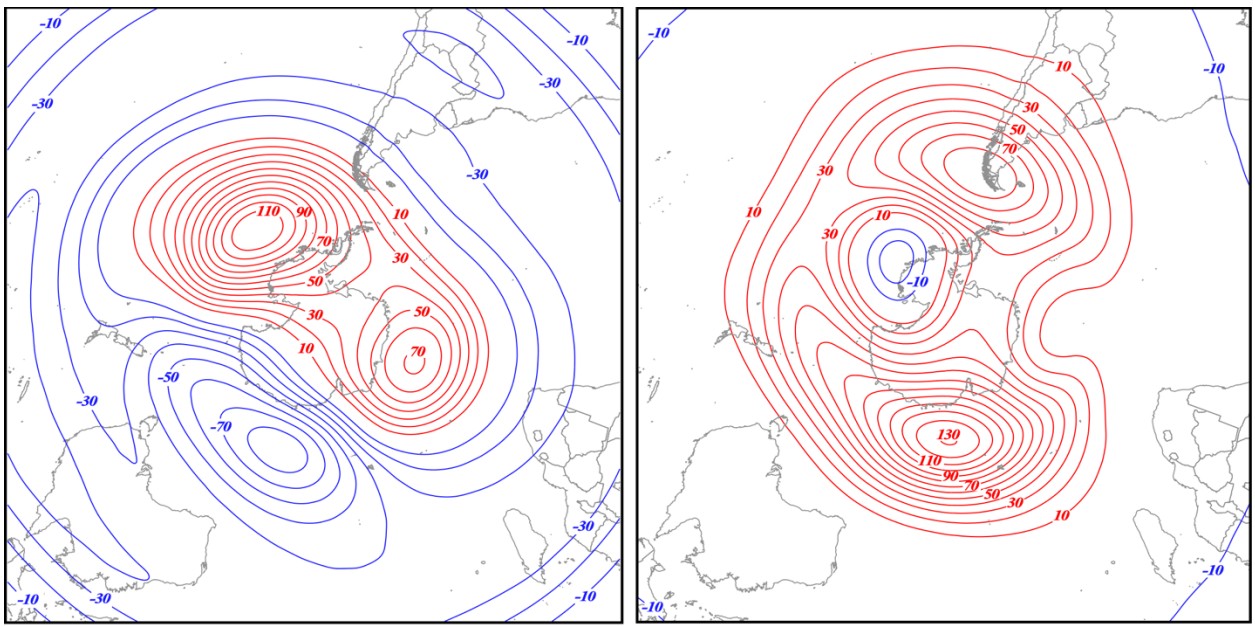

Fig. 13  50 hPa height differences between the composite waviest and least wavy (a) polar jet and  (b) subtropical jet seasons constructed from the JRA-55 reanalysis.  See Table 1 for identification of the specific years comprising each composite.  Positive (negative) height differences are in solid red (blue) lines labeled in m and contoured every 10 m (-10 m) beginning at 10 m (-10 m).
