# Peer review of "WAVINESS OF THE SOUTHERN HEMISPHERE WINTERTIME POLAR AND SUBTROPICAL JETS by Jonathan E. Martin1 and Taylor Norton2 1Department of Atmospheric and Oceanic Sciences 2Antarctic Meteorological Research Center University of Wisconsin-Madison Madison, WI 53706"

_EGUsphere, 2023_

## Author Response (AR1)

**Response to Reviewer 1**

General comments

This study investigates trends in the waviness of the Southern Hemisphere jetstreams using a recently-developed metric, the averaged latitudinal displacement (ALD). While I believe the waviness of the Northern Hemisphere has been extensively studied, the Southern Hemisphere has been paid much less attention. As such, I believe the findings presented in this preprint are of sufficiently novel and important scientific value. While the analysis and discussion of tropospheric trends seem more complete, the part about troposphere-stratosphere coupling lacks depth in my opinion. I also believe that the manuscript would benefit from discussing more the climatological properties (in addition to the case studies) of the ALD to demonstrate its ability to capture key features of the SH circulation.

We thank the reviewer for carefully reading our manuscript and offering many constructive comments. We have endeavored to revise the manuscript through incorporation of nearly all of those comments. We detail our responses, and highlight revisions in the resubmitted manuscript, in the following responses.

Response to Major Comments:

The discussion about Fig. 3 is overly terse. Please tell the reader why this figure is important. How can we interpret it? Otherwise, what is the reason for showing it?

1) New Figure 4 (old Figure 3) presents the distribution of the core isertel values over the time series for each jet from each data set. The purpose of showing this distribution is to make the point that the two layers do have robust, identifiable peaks in their respective distributions that mirror those found in a similar analysis of NH wintertime jets. The figure can be interpreted as evidence that the PV-based categorization of the tropopause-level jets as physical features is a robust method as it returns very similar results in both hemispheres. Text has been added in the revised manuscript to better assist in interpretation of this result.

NEW TEXT: "Figure 4 portrays the frequency of occurrence of the core isertels in both the STJ and POLJ layers for each of the three time series. The STJ core isertels peak between -1.95 and -2.1 PVU across the three data sets. Considering all three data sets, 81.5% of all JJA days exhibit a core isertel between -1 and -3 PVU in the STJ layer. The POLJ distribution is shifted toward higher PV values. Overall, 74.8% of JJA days had a core isertel between -1 and -3 PVU in the POLJ layer. "

The section on the potential coupling between the troposphere and the stratosphere (L235) feels like an afterthought. The dynamical coupling between the two layers has been thoroughly studied for many decades in both hemispheres and much insight was

obtained on the role of vertical wave propagation and mean flow interactions. For instance, it is well understood how the location of transient eddies relative to the climatological flow can modulate wave propagation through linear interference. The metric proposed, the ALD, does not however offer information on the location of disturbances and cannot capture such a process. It neither offers insight into "waviness fluxes". There is also barely any literature cited on troposphere-stratosphere dynamical coupling (only one paper from 50 years ago). In its current form, the discussion offers little in terms of novelty while failing to properly acknowledge past scientific findings on the topic.

2) We agree that the short section regarding possible coupling between the troposphere and stratosphere is not substantial enough to be included in the analysis.  Though comparison of the waviest and least wavy years may be of interest, these ideas are not sufficiently well developed to take their place next to others that are more fully formed.  And you are absolutely correct in stating that ALD does not make distinctions among various waves, their locations, or their individual amplitudes.  It perhaps could be altered to allow such distinctions, but we have not considered that refinement to date.  We have dropped this short section in the revised manuscript and taken up a more substantial discussion of the relationship between our results and characteristics of the Southern Annular Mode (SAM) as suggested by you and another reviewer.

Figure 8  (left panel) reveals a certain degree of zonal symmetry. This, therefore, raises the question of whether the ALD is related to the Southern Annular Mode (SAM). I'd encourage the authors to look at the relationship between ALD and the SAM, a very well-documented mode of variability, rather than dabbling into troposphere-stratosphere coupling. For instance, I'd be curious to see spaghetti plots of core isertels for the negative and positive polarities of the SAM. Also, what is the correlation with the SAM index?

I also think that it would be very valuable to show more climatological properties of the ALD. Figure 2 shows nice illustrative cases of the isertel's alignment with the jets but does not show if this is generally the case. Would the climatological mean isertel align with the climatological wind maxima at 250 and 850 hPa? It would add value to show the typical locations of jets from the perspective of the mean position of isertels on maps.

3) We enthusiastically took up your suggestion of looking at any possible relationship between our ALD metric and characteristics of the SAM.  As one example, we considered the three winter months with the most positive and most negative SAM extremes since 1979 (June 2009, July 1998, and August 1994 for the former and June 1992, July 1995, and August 1981 for the former).  New Fig. 10 is a spaghetti plot of core isertels for each pair of winter months that illustrates the clear distinction between the extremes.  Positive (negative) extremes of SAM

show a poleward encroachment (equatorward displacement) of the SH polar jet.  In June, the POLJ waviness is larger in the POS SAM extreme but that relationship is reversed in both July and August when the POS SAM extremes are less wavy than their NEG SAM counterparts.

We also considered the JJA 3-month average SAM index (calculated after Gong and Wang (1999)) to the winter ALD from the JRA-55 reanalysis (new Fig. 9).  The correlation between the two time series was 0.053 suggesting almost no relationship between SAM and the seasonal averaged ALD.  Both of these analyses are presented with new figures in the revised manuscript.

Finally, we calculated the average latitude of the core isertel for both the STJ and POLJs from each data set.  Based upon the hemispheric distribution of these average core isertels, we have calculated the climatological ALD for both jets and report that now in the revised text.  These average core isertels, along with isotachs of the 200 hPa jet (for the STJ) and 300 hPa jet (for the POLJ) are shown in new Fig.  3 and described in the accompanying text.  The core of the STJ isotach maxima around the hemisphere is a near-perfect fit for the nearly indistinguishable core isertels of the STJ.  The fit between the core isertels of the POLJ and the 300 hPa isotachs is not quite as stunning, but is convincing nonetheless.

NEW TEXT: Lines 87:108 and 199:205 of revised manuscript

A rather similar metric, sinuosity, was previously employed to diagnose waviness in the Northern Hemisphere (Cattiaux et al., 2016). Differences between the two should be explained.

4) Application of the sinuosity metric to the jet stream was my original idea and yet, for various reasons, I was not included as an author on that Cattiaux et al. (2016) paper to which you referred.  Sinuosity in this context suffers an incurable resolution dependence, since the calculation depends entirely on measuring contour length. Such dependence is at the heart of the famous "coastline" problem of Richardson in which measuring the length of the coastline of England depends entirely on the desired resolution.  ALD sidesteps part of this problem by only depending once on contour length (in the calculation of circulation that leads to identification of the core isertel).  After that core isertel and its equivalent latitude are identified, ALD simply measures the latitudinal displacement of the core isertel from the equivalent latitude.  This step is NOT resolution dependent.  In addition, the ALD method is premised on dynamical identificadtion of a physically relevant, feature-based characteristic of the tropopause-level flow - the region of strongest horizontal PV gradient. With sinuosity one has to assume that a set of geopotential height contours at a given level are well chosen to describe the feature of interest – and one is restricted to consideration of the waviness of the geostrophic flow, not the total flow.  My sense is that ALD is superior to sinuosity as a metric of waviness for these reasons.

Minor Comments:

L35: "The analysis reveals both similarities and differences" This sentence is very vague. Not very helpful for the reader.

1) You are right about this – the original intention of this sentence was to set the reader up for the sentence that follows, but it is not needed.  We have removed it without any ill effect on the message conveyed in the abstract.

L45: "negatively impacted". This judgment may not be obvious to all readers. Better be more descriptive and use "weakened" or "decelerated" here.

2) This is a very good suggestion but that portion of the text has been removed in the revised manuscript.

L189: I do not understand what the 93% means here.

3) Using the three data sets of varying lengths (72 years for NCEP, 62 years for JRA-55, and 41 years for ERA-5) we have 175 annual time series of the ALD of both jets.  This statement means that 163 of those pairs are correlated with magnitudes less than 0.3.

**Response to Reviewer 2:**

The paper presents an assessment of the waviness of SH subtropical and eddy-driven jets in three reanalyses. The paper is well written and the results are sound and clearly expressed. The long-term trends show an increasing waviness of the SH jets, together with a poleward shift and no change in the strength of the jet. I have the following suggestions before publication.

We thank the reviewer for carefully and thoughtfully reading our manuscript and for offering a number of excellent suggestions for its revision. We have attempted to incorporate all of the suggestions into the revised manuscript which stands much improved as a result. We offer our point-by-point responses below.

Minor Comments:

L124-133: How do your results compare with other jet definitions or algorithms in the literature? (e.g. Manney, G. L., Hegglin, M. I., Lawrence, Z. D., Wargan, K., Millán, L. F., Schwartz, M. J., Santee, M. L., Lambert, A., Pawson, S., Knosp, B. W., Fuller, R. A., and Daffer, W. H.: Reanalysis comparisons of upper tropospheric–lower stratospheric jets and multiple tropopauses, Atmos. Chem. Phys., 17, 11541–11566, https://doi.org/10.5194/acp-17-11541-2017, 2017.

Manney, G. L., and M. I. Hegglin, 2018: Seasonal and Regional Variations of Long-Term Changes in Upper-Tropospheric Jets from Reanalyses.*J. Climate*, **31**, 423–448, https://doi.org/10.1175/JCLI-D-17-0303.1.)

1) Manney et al. (2017) and Manney and Hegglin (2018) have done excellent work on identifying the polar and subtropical jets and investigating the trends in their various characteristics over the past decades. Their approach identifies the jets diagnostically as volumes within which the wind speed is >= 40 m s$^{-1}$. Our work is premised on having identified isentropic layers wherein the wind speed maxima (of greater than 30 m s-1) is located. This allows our method to hunt for dynamical signals (i.e. the horizontal PV gradient) that identify the core of the two jets. Manney's work is more diagnostic and employs latitude criteria for differentiating between POLJ and STJ that include well considered definitions based upon the tropopause height, the latitude of the jet maxima, and other considerations. They also celebrate the agreement amongst the different reanalysis data sets as evidence of the robustness of their results as we do here. Though not presented in this paper, Martin (2021) found that the NH POLJ is shifting poleward in direct contradiction to the result from Manney and Hegglin (2018). Our results in this paper bear a closer correspondence to those in Manney and Hegglin (2018) with both the SH POLJ and STJ shifting poleward and both studies note an increase in the STJ core speed as well. We have added reference to these differences and similarities to the Discussion section of the revised paper.

NEW TEXT: . "The use of isentropic space here differs from the insightful approach taken by Manney et al. (2017) and Manney and Hegglin (2018) which employed separate latitude and elevation criteria to differentiate between the STJ and the POLJ."

L182-185: This is the trend of the reanalysis average, right? It would be good to discuss the trends in the individual reanalyses, to check the level of agreement.

2) The trend discussed in relation to new Fig. 5 is the trend from the JRA-55 data, chosen as representative. The revised text includes mention of the trends from the other two data sets for completeness and they are – NCEP POLJ (from 1958), 0.023 deg/year; NCEP STJ (from 1958), 0.0125 deg/year; ERA5 POLJ, 0.0088 deg/year, and ERA5 STJ, -0.001/deg/year). The figure in the revised manuscript does not include a separate trend line for each reanalysis time series.

L211-212: The poleward migration of the SH jets has been linked to the ozone hole (see for example WMO Scientific Assessment of Ozone Depletion: 2022, Chapter 5. Stratospheric Ozone and Climate, https://csl.noaa.gov/assessments/ozone/2022/downloads/Chapter5_2022OzoneAssessment.pdf, and references therein).

3) We have added a reference to the suggested connection between poleward migration of the SH jets and the ozone hole as suggested. Thank you for pointing this possible connection out to us.

L243-244: It is not such intriguing implication, as the stratospheric polar vortex is perturbed by the upward propagating waves originated in the troposphere. In order to examine this connection you could look at the eddy heat flux at 100 hPa averaged over some extratropical latitudinal range (something like 45-75 S), as it is indicative of the amount of upward wave propagation that can then disturb the polar vortex.

4) This is an excellent suggestion but, as both other reviewers were also critical of the lack of substance in the very short troposphere/stratosphere section of our analysis, we have decided to remove any reference to possible dynamical explanations from this manuscript and take it up in separate work that will require development of some new expertise on our parts.

Technical:

L36: use 'trends' instead of 'tendencies', as the latter typically refers to short time variability?

1) We have removed that sentence from the abstract per a suggestion from Reviewer 1 – but have combed through the text of the revised manuscript to ensure that we use "trends" rather than "tendencies" as you suggest here.

L45: change 'negatively impacted' by 'weakened', it is clearer and more consistent with literature terminology

2 This is a very good suggestion but that portion of the text has been removed in the revised manuscript.

L134-137: It is unclear what you mean with this sentence, please clarify.

3) This sentence has been rewritten to clarify – it now says "It is important to note that 53.8% of all qualifying columns (to 380K) in the 0-40S bin (STJ) were in the 340-355K layer while 46.8% of all qualifying columns in the 40-65S bin (POLJ) were in the 310-325K layer *offering strong support for the isentropic assignments for the two species mentioned previously*."

L41: Change poleward 'creep' by poleward 'shift'

4) We have made the suggested change.

**Response to Reviewer 3**

Martin and Norton investigate the waviness, strength, and position of the austral polar and subtropical jetstreams using their "core isertel" and averaged latitudinal displacement (ALD) metrics. They compute trends in these quantities and compare these between the polar and subtropical jets, and with those computed from the Northern Hemisphere. This paper is relatively novel in that the Southern Hemisphere jets have generally received much less attention in this regard than their Northern Hemisphere counterparts. However, there are a few issues with the manuscript that I think should be addressed before it is suitable for publication.

We thank the reviewer for carefully and thoughtfully reading our manuscript and for offering a number of excellent suggestions for its revision. We have attempted to incorporate all of the suggestions into the revised manuscript which stands much improved as a result. We offer our point-by-point responses below.

Major Comments:

While I appreciate the brevity of the paper, it also (in my opinion) sorely lacks necessary discussion and ties with other relevant literature. For instance, the introduction focuses quite specifically on jet metrics and the reference to Gallego et al (2005) being the "only one to consider aspects of waviness". However, this ignores a large body of literature on the Southern Hemisphere circulation with closely related aspects such as the Southern Annular Mode (SAM), the jet/circulation response to the ozone hole and recovery, etc. In fact, there are quite a few recent papers that have focused on the prominent zonal wavenumber 3 pattern in the SH circulation (e.g., papers by R. Goyal et al in Nature Geosci and JClim, and E. Campitella in ClimDyn). The results shown/discussed in this paper do not exist in isolation from other relevant research. Furthermore the value/novelty of the paper could be increased by examining these jet metrics in the context of, e.g., the SAM (I note this is also mentioned by one of the other reviewers).

1) We enthusiastically took up your suggestion of looking at any possible relationship between our ALD metric and characteristics of the SAM. As one example, we considered the three winter months with the most positive and most negative SAM extremes since 1979 (June 2009, July 1998, and August 1994 for the former and June 1992, July 1995, and August 1981 for the former). New Fig. 10 is a spaghetti plot of core isertels for each pair of winter months that illustrates the clear distinction between the extremes. Positive (negative) extremes of SAM show a poleward encroachment (equatorward displacement) of the SH polar jet. In June, the POLJ waviness is larger in the POS SAM extreme but that relationship is reversed in both July and August when the POS SAM extremes are less wavy than their NEG SAM counterparts.

We also considered the JJA 3-month average SAM index (calculated after Gong and Wang (1999)) to the winter ALD from the JRA-55 reanalysis (new Fig. 9). The correlation between the two time series was 0.053 suggesting almost no relationship between SAM and the seasonal averaged ALD. Both of these analyses are presented with new figures in the revised manuscript.

Finally, we calculated the average latitude of the core isertel for both the STJ and POLJs from each data set. Based upon the hemispheric distribution of these average core isertels, we have calculated the climatological ALD for both jets and report that now in the revised text. These average core isertels, along with isotachs of the 200 hPa jet (for the STJ) and 300 hPa jet (for the POLJ) are shown in new Fig. 3 and described in the accompanying text. The core of the STJ isotach maxima around the hemisphere is a near-perfect fit for the nearly indistinguishable core isertels of the STJ. The fit between the core isertels of the POLJ and the 300 hPa isotachs is not quite as stunning, but is convincing nonetheless.

NEW TEXT: Lines 87:108 and 199:205 of revised manuscript

I also agree with the other reviewers that the analysis presented related to coupling with the stratospheric polar vortex is not very compelling or intriguing. The ALD metric can't tell us anything about phasing and amplification of waves with the climatological stationary wave pattern, but it's still a reasonable expectation that the most extreme wavy cold seasons would correspond to somewhat weaker polar vortex conditions.

2) We agree that the short section regarding possible coupling between the troposphere and stratosphere is not substantial enough to be included in the analysis. Though comparison of the waviest and least wavy years may be of interest, these ideas are not sufficiently well developed to take their place next to others that are more fully formed. And you are absolutely correct in stating that ALD does not make distinctions among various waves, their locations, or their individual amplitudes. It perhaps could be altered to allow such distinctions, but we have not considered that refinement to date. We have dropped this short section in the revised manuscript and taken up a more substantial discussion of the relationship between our results and characteristics of the Southern Annular Mode (SAM) as suggested by you and another reviewer.

I would like to see a more rigorous handling of the various trend analyses presented. There are much better/robust statistical techniques than t-tests for evaluating the significance of trends (e.g., examining 15-20 year running trends across the full time series to better understand decadal-scale variability and sensitivity to time series endpoints; bootstrapping; etc). This is all the more important since the reanalysis data are likely to be of somewhat dubious quality in the pre-satellite era Southern Hemisphere (see, e.g., Hitchcock 2019)

3) We have added additional details on trends from the various data sets in the revised text.

Minor Comments:

Along the same lines as my last major comment, I really don't think including the NCEP/NCAR reanalysis is of any value, as it is a very old/obsolete reanalysis. I won't complain if the authors prefer to keep it in, but at some point I think the research community should actively discourage its use in research (hence why I say this in an open access review).

1) We will keep the NCAR-NCEP reanalysis in the study as it provides a symmetry with the Martin (2021) study of the wintertime NH tropopause-level jets.

ERA5 now has a back-extension to 1940, and thus it could be compared with JRA55 for 1958-forward. However, I also understand obtaining and analyzing more of such a large dataset could be prohibitive.

2) As you surmised, getting the ERA-5 data back to 1940 is too onerous for the short period allotted for response to reviewers comments in this case, but we look forward to acquiring that data set for future analyses.

Specific Comments:

L203: Please clarify -- this sounds like it deviates from the standard definition of equivalent latitude, which would be the latitude circle with the same area as that enclosed poleward of the core isertel. This statement sounds as if you average the latitude of every point on the isertel contour around all longitudes.

1) This is an excellent correction – the EQLAT very nearly approximates the zonally averaged latitude but they are not the same thing. We have changed the wording here to state " . . the jet core's equivalent latitude which closely approximates its zonally averaged position."

L216-221 and Table 1: Why do you leave this as an integrated sum rather than just taking the average so that it's the seasonally averaged ALD anomaly? It seems that would be more intuitive to understand the numbers in the table.

2) Since there is no substantive difference between what we have done and what you suggest, and that the suggestion does not make interpretation of the various numbers any more readily available, we have opted to not make a change here.

Fig 4: While I understand the NH results are included for comparison/context, including each grey line from each reanalysis makes the figure very noisy. This particularly has a negative impact on the ability to examine the POLJ results. I suggest using a multi-reanalysis mean for 1958 forward so these results can be reduced to a single grey line for the NH.

3) We have made this amendment, an excellent suggestion, to old Figs. 4, 6, and 7 (new Figs. 5, 7, and 8).

Fig 8 and 9: I want to echo Referee #1's comment here regarding the SAM. I also note that Fig 8b and 9b both prominently show a wave-3 pattern (see my first major comment, and references herein).

4) Addition of new analyses comparing ALD to the SAM, in other parts of the paper, allow us to comment only briefly on the apparent wave-3 look to these composite difference fields. We do make those comments in the revised text.

NEW TEXT: Lines 297 and 304 in revised text.